# Relationship of CT densitometry to lung physiological parameters and health status in alpha-1 antitrypsin deficiency: initial report of a centralised database of the NIHR rare diseases translational research collaborative

Diana Crossley,[1] James Stockley,[2] Charlotte E Bolton,[3] Nicholas S Hopkinson,[4] Ravi Mahadeva,[5] Michael Steiner,[6] Tom Wilkinson,[7] John R Hurst  ,[8] Bibek Gooptu,[6,9] Robert A Stockley  [2]

For numbered affiliations see end of article.

**Correspondence to**
Dr Robert A Stockley;
r.a.stockley@bham.ac.uk

## ABSTRACT

**Objectives** To establish a database network for the study of alpha-1 antitrypsin deficiency (AATD) and compare the results to CT lung density as the most direct measure of emphysema.

**Design** A central electronic database was established to permit the upload of anonymised patient data from remote sites. Prospectively collected CT data were recorded onto disc, anonymised, analysed at the coordinating centre and compared with the clinical features of the disease.

**Setting** Tertiary referral centres with expertise in the management of AATD focused on academic Biomedical Research Units and Wellcome Clinical Research Facilities.

**Participants** Data were collected from 187 patients over 1 year from eight UK academic sites. This included patient demographics, postbronchodilator physiology, health status and CT. Analysis was undertaken at the coordinating centre in Birmingham.

**Results** Patient recruitment in the 12 months reached 94% of target (set at 200) covering the whole spectrum of the disease from those with normal lung function to very severe chronic obstructive lung disease. CT scan suitable for analysis was available from 147 (79%) of the patients. CT density, analysed as the threshold for the lowest 15% of lung voxels, showed statistically significant relationships with the objective physiological parameters of lung function as determined by spirometric Global Initiative for Chronic Obstructive Lung Disease (GOLD) severity staging (p<0.001) and carbon monoxide gas transfer (p<0.01). Density also correlated with subjective measures of quality of life (p=0.02).

**Conclusions** Establishment of the network for data collection and its transfer was highly successful facilitating future collaboration for the study of this rare disease and its management. CT densitometry correlated well with the objective clinical features of the disease supporting its role as the specific marker of the associated emphysema and its severity. Correlations with subjective measures of health, however, were generally weak indicating other factors play a role.

### Strengths and limitations of this study

► The strengths of the project have been the design and delivery of a centralised registry for alpha-1 antitrypsin deficiency enabling the whole spectrum of the disease to be characterised, especially including those with more severe disease who are less able to travel long distances to single National Centres of Excellence.

► Recruitment and data collection were rapid and 94% of the target.

► The study provided further validation of the specificity of quantitative analysis of lung density for the assessment of emphysema in this 'rare' disease.

► The limitations of the study included the slight shortfall in patient recruitment over the 1-year target and slightly incomplete postbronchodilator physiological data entry.

► A reduced number of CT scans were analysed because of recent routine scans performed under different parameters. Although all patients filled in one of the health status tools, this was not 100% for both.

## INTRODUCTION

Alpha-1 antitrypsin deficiency (AATD) affects around 1 in 2500 individuals in the UK and increases the susceptibility to develop chronic obstructive pulmonary disease (COPD) with an early-onset emphysema dominant phenotype and adult-onset liver cirrhosis. Alpha-1 antitrypsin (AAT) is the main plasma serine proteinase inhibitor and is predominantly made and secreted by the liver. It enters the tissues by simple diffusion especially in the presence of inflammation where it regulates the local activity of neutrophil proteinases.[1]

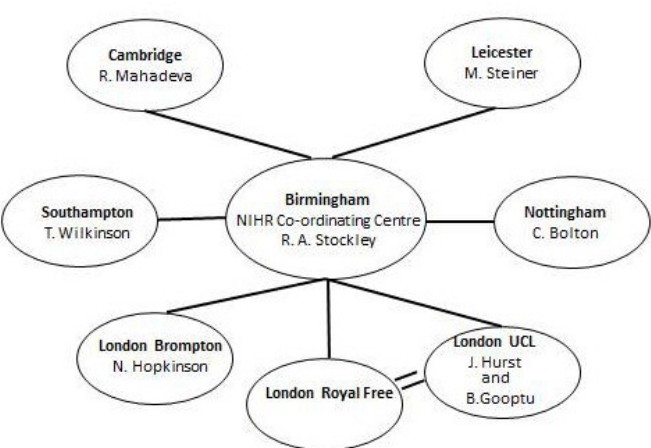

**Figure 1** The National Institute for Health Research rare diseases alpha-1 antitrypsin deficiency network individual centres are shown together with their principal Investigators and link to the coordinating centre in Birmingham.

This is important in delicate tissue structures such as the alveolar region of the lung where uncontrolled serine proteinases can destroy connective tissues leading to alveolar destruction and the development of emphysema.[2]

The study and treatment of rare diseases present many problems including patient identification, referral to centres with relevant expertise, consistent demographic

| Table 1 Baseline characteristics table | | |
|---|---|---|
| **Clinical variable** | **Number of subjects** | **Median (IQR)** |
| Age (years) | 187 | 60.1 (50.9–66.9) |
| Sex | | 86M 101F (46%M, 54% F) |
| Pack years for ex-smokers | 99 | 17 (8–27) |
| FEV$_1$ (L) | 185 | 1.4 (0.9–1.9) |
| FEV$_1$ per cent predicted | 185 | 51.1 (30.5–67.5) |
| FVC (L) | 185 | 3.6 (2.9–4.8) |
| FVC per cent predicted | 185 | 105.7 (88.2–123.4) |
| FEV$_1$/FVC (%) | 185 | 36.9 (28.5–49.8) |
| Kco per cent predicted | 181 | 56.3 (46.0–69.6) |
| RV/TLC (%) | 140 | 41.5 (33.0–52.7) |
| Voxel index −950 HU (%) | 147 | 24.5 (15.4–35.2) |
| Perc15 (HU) | 147 | −965.2 (−974.8 to −950.7) |
| Total CAT score | 158 | 18.9 (12.0–25.0) |
| Total SGRQ score | 158 | 45.2 (33.3–62.1) |

Data are shown as median and interquartile ranges together with the number of patients where data were collected. All subjects were either never smokers or had ceased at least 1 year before the time of assessment. CAT score is only given for those where the total SGRQ score was also available.
CAT, COPD assessment tool; FEV$_1$, forced expiratory volume in 1 s; FVC, forced vital capacity; HU, hounsfield units; Kco, carbon monoxide transfer coefficient; RV/TLC, residual volume/total lung capacity; SGRQ, St George's Respiratory Questionnaire.

characterisation and importantly the design and delivery of appropriately powered research studies and clinical trials. With this in mind, the UK National Institute for Health Research (NIHR) commissioned a series of projects in 2014 to develop networks for the study and deep phenotyping of rare diseases with the longer term aim of providing a consistent structure to facilitate collaboration between groups and with industrial partners for research and therapeutic development.

In 1996, the Antitrypsin Deficiency Assessment and Programme for Treatment (ADAPT) was established in Birmingham as an investigator led, industry and research grants funded project in order to study AATD and its natural history. The programme depended on referrals, mainly from secondary care centres in the UK. Patients were invited to visit Birmingham on an annual basis for assessment as part of a clinical/research programme (Local Research Ethics Committee (LREC) 3359a). However, although successfully collating data on over 1000 deficient patients, those seen may be affected by bias due to referral patterns as well as to issues of geography and transport difficulties for annual visits. The latter issues may be especially relevant for the most severely affected patients. The ADAPT cohort may therefore underestimate the true impact, especially of the severe end of the disease spectrum. In addition, the evolution of a single centre impairs delivery of near patient care, clinical trials and iterative thought, which can be handicapped by the absence of other local centres of excellence more convenient for patient attendance, management and, especially, recruitment/participation in research and clinical trials.

The current article describes the AATD collaborative network funded by the NIHR to establish a patient cohort for rare diseases collated over 1 year in the UK. In particular, we wished to recruit patients with a wide range of severity, with the most severe being recruited to the closest specialist centre to minimise the issue of difficulty of travel. In particular, we wished to assess lung densitometry as the most direct measure of emphysema and physiological impairment and patients perception of their health status.

## MATERIALS AND METHODS

Birmingham acted as the coordinating centre for the NIHR Rare Diseases Translational Research Collaboration in AATD. Links were made with academic biomedical research units where a specialist with expertise in clinical trials and interest in AATD had been identified. The network had an initial target of 200 patients with AATD and the PiZZ genotype recruited from Birmingham, Nottingham, London Brompton, London University College London/Royal Free Hospital, Cambridge, Southampton and Leicester (see figure 1). The aim was to recruit 100 patients from Birmingham with mild and moderate disease and 100 patients with severe and very severe disease (as defined by the forced expiratory

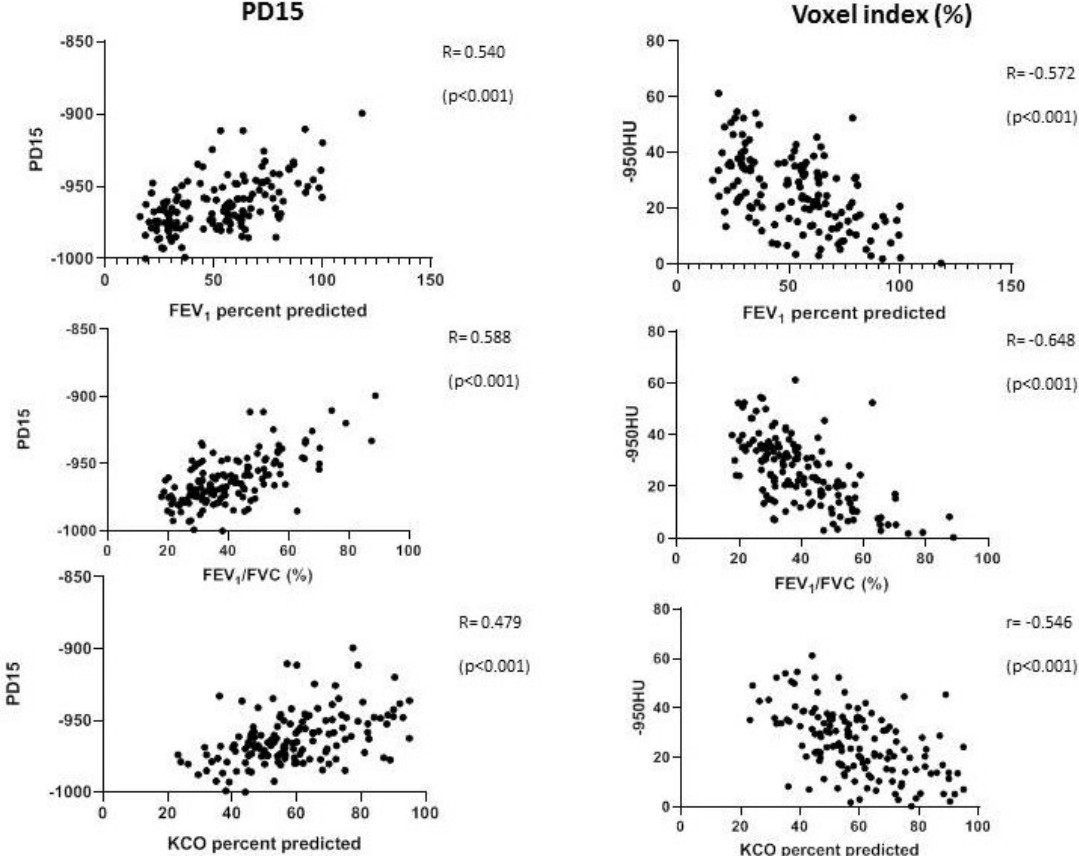

**Figure 2** Scatter plots of the relation between lung function and CT densitometry analysis. Each point represents data from a single patient. The correlation coefficient (r) is given for each analysis using the two best-recognised parameters for emphysema on CT scan. KCO, carbon monoxide transfer coefficient; FVC, forced vital capacity; FEV$_1$, forced expiratory volume in 1 s.

volume in 1 s (FEV$_1$)) from the other centres such that the distance these patients had to travel was less than attending a single central facility. A standard protocol was provided to all centres based on that used in all previous publications from the core Birmingham centre and consisted of general demographic data required for, in depth, patient characterisation (the lung function equipment used for the study varied between centres and is summarised for each site in the online supplement). This characterisation would also provide the basis of patient identification for inclusion in subsequent clinical trials using national standard operating procedures for lung function and specific parameters for CT acquisition.

Patients were recruited sequentially from the individual centre clinics according to the spirometric criteria outlined above.

All subjects underwent full clinical examination were scheduled to undergo full postbronchodilator lung function testing (according to the Association of Respiratory Technology/British Thoracic Society guidelines for quality control) including spirometry gas transfer and lung volumes, determination of current health status using well-established tools (St George's Respiratory Questionnaire (SGRQ) and the COPD assessment tool (CAT)). Where routine high resolution CT had not been undertaken within the previous 2 years, a scan on full

inspiration was taken using the following fixed criteria: subjects were scanned by spiral multislice CT of the chest in the supine position within 4 hours of administration of a short-acting bronchodilator. A low radiation dose (140 kVp) was used, the slice thickness was 5 mm and increments of 2.5 mm, and a soft reconstruction algorithm (B30f) was applied. The anonymised data were subsequently transferred to Birmingham on disc for quantitative analysis using Pulmo-CMS (Medis Medical Imaging, Leiden, The Netherlands courtesy of Berend Stoel). Measurements for voxel index (% voxels less dense than −950 HU) and Perc15 (density threshold of the lowest 15% of the voxels) were assessed for the whole lung such that greater voxel index or lower Perc (PD)15 indicated a greater amount of emphysema.

### Patient and public involvement

The project was approved by five patients who completed a traffic light system research proposal review form to give their opinion on the research project and the significance of its impact on clinical practice. A further patient representative was a formal member of the steering committee. The information will be part of future presentation to the patient group and summarised for dissemination in their annual newsletter once accepted for publication. Results will also be presented at international speciality meetings.

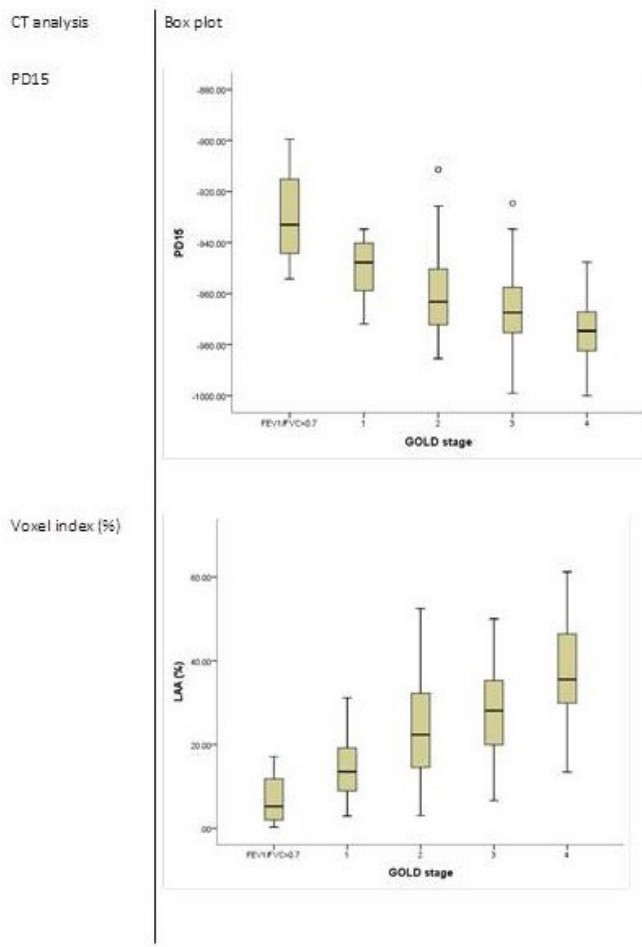

**Figure 3** Box plots of CT parameters by GOLD stage. Data are shown as box plot of IQR with median value indicated by the solid line, whiskers indicate 95% data range and outliers are indicated by open circles for each GOLD stage of severity. LAA, low area of attenuation.

**Table 2** Multivariate linear regression models between CT density and lung function

| CT density | Lung function parameter | Unstandardised coefficient | SE |
|---|---|---|---|
| PD15 | FEV$_1$pp | 0.63 | 0.1 (p<0.001) |
| | FEV$_1$/FVC (%) | 0.94 | 0.1 (p<0.001) |
| | Kcopp | 0.55 | 0.1 (p<0.001) |
| Voxel index (%) | FEV$_1$pp | −0.48 | 0.1 (p<0.001) |
| | FEV$_1$/FVC (%) | −0.67 | 0.1 (p<0.001) |
| | Kcopp | −0.5 | 0.8 (p<0.001) |

Data are shown for the regression analysis of both CT parameters and baseline lung function. 'pp' is the per cent predicted.
FEV$_1$, forced expiratory volume in 1 s; FVC, forced vital capacity.

This project represented a first step towards achieving the long-term aims expressed by patients as part of the European Lung Foundation survey of AATD.

## Statistical analysis

Baseline characteristics are presented as median and IQR. The Pearson's or Spearman rank correlation coefficient between CT density and lung function was calculated. Analysis of variance or Kruskall-Wallis was used to assess differences between grouped data. Normality of the data was tested using the Shapiro-Wilk test, and the level of significance was conventionally determined at p<0.05.

## RESULTS
### Baseline characteristics

There were 187 patients recruited to the study across the eight sites within the 1-year recruitment time. Of these, 84 with severe and very severe COPD were recruited from the satellite sites and 94 mild and moderate disease (as well as 9 local patients with very severe disease) were recruited by Birmingham. Confirmed post bronchodilator spirometry was available for 185 patients and carbon monoxide gas transfer (single breath) for 181. Quantitative CT scan was obtained for 147 patients and all patients completed quality of life (QoL) metrics (SGRQ and/or CAT) although only 158 patients completed both.

The baseline characteristics are summarised in table 1. Data are presented as median and IQR to characterise the spread of patients studied. The median age was 60 years and the median postbronchodilator FEV$_1$ for the cohort was 51.1% predicted for age, sex, height and ethnicity.[3] Of these, 44 (24%) patients (recruited from the peripheral sites) were graded as Global Initiative for Chronic Obstructive Lung Disease (GOLD) stage 4 at the time of study.[4] Nine patients with AATD (5%) had an FEV$_1$/forced vital capacity (FVC) >0.7. The median CAT score for the whole group was 18.9 (12.0–25.0) and the median total SGRQ score was 45.2 (33.3–62.1).

Smoking history information was available for 178 patients. Seventy-nine were lifelong never smokers but all patients with a smoking history had ceased for more than 12 months at the time of recruitment. The median smoking history for the ex-smokers (n=99) was 17 pack years (IQR 8–27). None of the patients was receiving AAT augmentation therapy as it remains unavailable for management of emphysema in the UK.

### Cross-sectional correlations
#### Lung function and CT densitometry

The relationship between CT density and lung function is shown in figure 2. There were significant correlations between both lung density parameters and the physiological parameters shown (p<0.001 all comparisons). Significant differences were also seen for CT density measurements as both the voxel index (which quantifies the proportion of low-density voxels below −950 HU) and PD15 (which quantifies the threshold below which

**Table 3** Quality of life data across each GOLD stage[4]

| GOLD stage | Patient number | CAT median (IQR) | Patient number | SGRQ median (IQR) |
| --- | --- | --- | --- | --- |
| FEV1/FVC>0.7 | 9 | 12 (6–21) | 9 | 27.1 (9.7–48.0) |
| 1 | 13 | 15 (9–21) | 12 | 33.4 (19.3–35.7) |
| 2 | 63 | 16 (12–21) | 63 | 41.5 (30.9–55.4) |
| 3 | 39 | 19 (11–25) | 39 | 48.8 (36.5–67.3) |
| 4 | 53 | 20 (13.5–27) | 33 | 62.0 (36.5–67.3) |

The number of patients for each Global Strategy for Obstructive Lung Disease (GOLD) stage is shown together with the median and IQR data for the CAT and SGRQ total score.

CAT, COPD assessment tool; $FEV_1$, forced expiratory volume in 1 s; FVC, forced vital capacity; SGRQ, St George's Respiratory Questionnaire.

15% of the overall voxels are sited), between each of the GOLD severity stages (p<0.001) (figure 3). There was no correlation between residual volume/total lung capacity (RV/TLC) and any of the densitometry parameters.

Following adjustment of all these relationships for age, sex and pack years, the relationship between density as measured by PD15 and voxel index remained significant (p<0.001) with no evidence of collinearity (see table 2).

### QoL measures and lung function

Within the cohort, the CAT test scores ranged between 1 and 39/40 (median=18.9; IQR 12–25) and related to disease severity as defined by the GOLD stage and summarised in figure 4 (p<0.0001). The median SGRQ was 45.2 (33.3–62.1) and also related to the GOLD stage (p<0.001) shown in figure 4 (p<0.001). The median and IQR for CAT and SGRQ in each GOLD stage are summarised in table 3.

There were significant correlations (online supplementary table 1) between both QoL measures and spirometry, as measured by $FEV_1$ (per cent predicted), FVC (per cent predicted), $FEV_1$/FVC (%) and with gas transfer coefficient, Kco (per cent predicted) and gas trapping as measured by RV/TLC % (p<0.01 all comparisons). Online supplementary figures 1 and 2 summarise the correlations for SGRQ and CAT with $FEV_1$ (% predicted).

### QoL measures and CT density

Both QoL measures were worse the greater the emphysema as quantified by the lung density. There was a significant relationship between voxel index and CAT (r=0.18, p=0.019) and a weaker relationship with PD15 (r=0.15, p=0.043) (figure 5). Total SGRQ correlated significantly with both measures of CT density, although again the relationship was weak ($r^2 <0.1$) (see figure 5).

### DISCUSSION

The current paper reports demographic data collated from a unique NIHR funded project to establish collaborative cohorts of patients with rare diseases focused on academic Biomedical Research Units and Wellcome Clinical Research Facilities. The main purpose of this project was to recruit 200 patients (none of whom were receiving augmentation therapy) with AATD (PiZZ genotype)

to a central database over a 1-year period as a basis for ongoing collaborative research. The aim of the collaborative was to recruit and characterise patients with AATD in depth across centres with a wide spectrum of severity. Importantly we wished to include those with severe and very severe disease (GOLD stages 3 and 4) as quantified by baseline $FEV_1$ (4) in the satellite centres local to their dwelling, who are more likely to refrain from travel to a

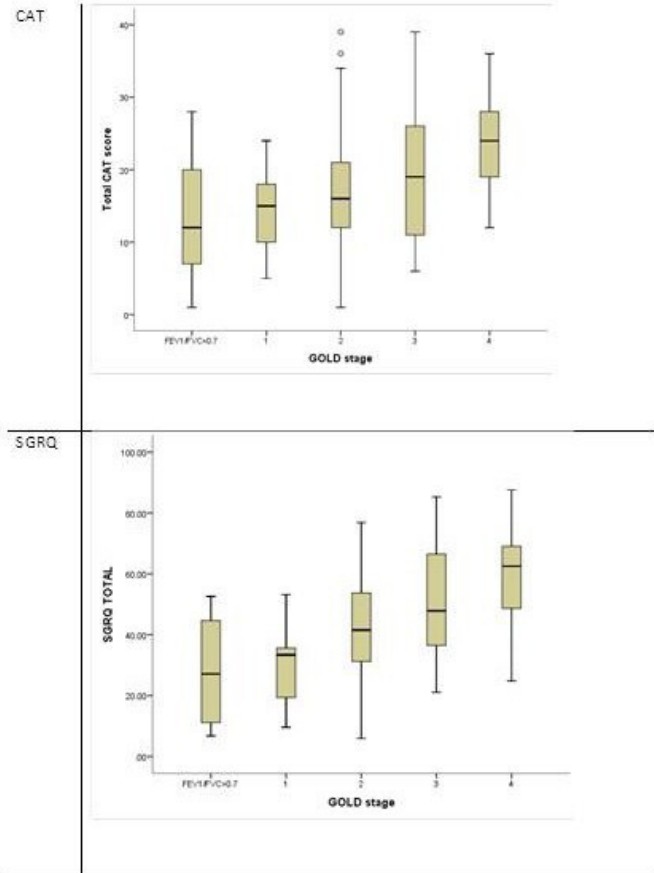

**Figure 4** Quality of life parameters determined by GOLD stage. Data are shown as box plot of IQR with horizontal median line, whiskers indicate 95% data range and outliers are shown as open circles. Patients with no airflow obstruction are shown as a separate group. CAT, COPD assessment tool; SGRQ, St George's Respiratory Questionnaire.

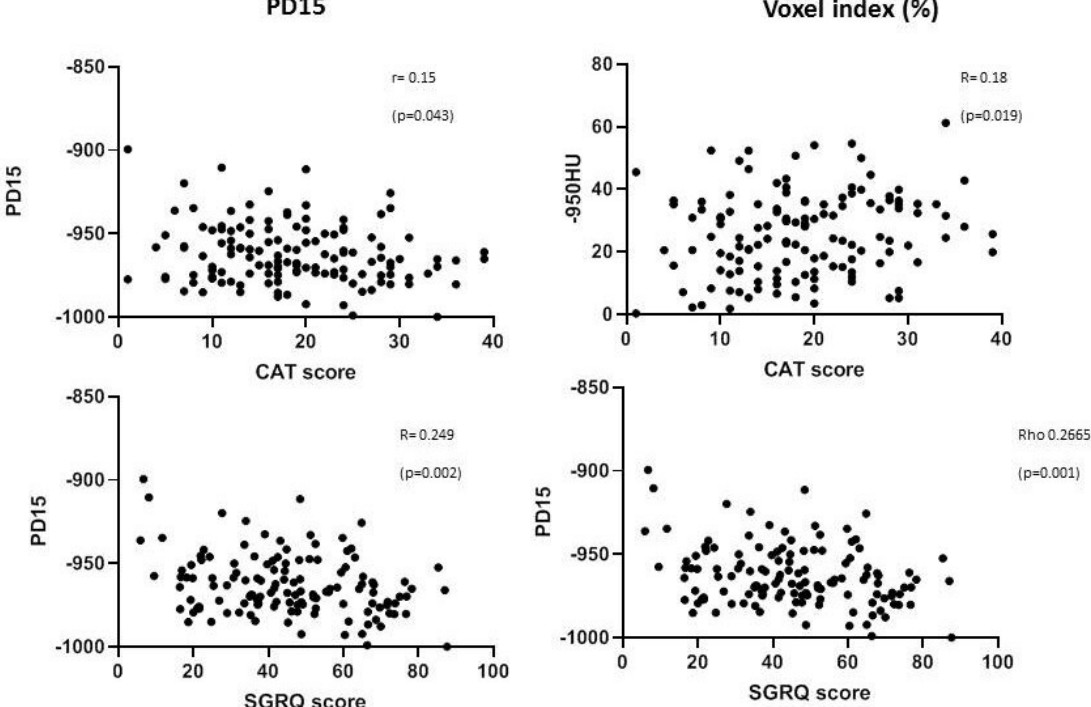

**Figure 5** Scatter plots to show the relation between quality of life and CT analysis. Data are shown for individual patient CAT and total SGRQ scores related to emphysema parameters. The significance of the relationships is indicated (p).CAT, COPD assessment tool; SGRQ, St George's Respiratory Questionnaire.

more distant central facility. Of note the data for the very severe patients produced similar values and demographic relationships to those seen historically in the single centre ADAPT programme in Birmingham.

We have confirmed the feasibility of this approach with recruitment from multiple academic centres in this limited time period reaching 94% of the target. Of these 98% had confirmed postbronchodilator spirometry (in 2 patients it had not been documented), 92% were able to complete gas transfer measurements and 85% had both the SGRQ and CAT measurement of health status although most patients (95%) had CAT recorded. CT scans were a unique feature of the UK patients and were performed prospectively at baseline using a fixed protocol and copied anonymously on to disc for analysis in Birmingham. This resulted in a total of 147 scans available and suitable for analysis (79% of patients) enabling the relationships to clinical parameters to be explored.

The patients showed a wide spectrum of disease including 53 patients local to the participating centres classified as GOLD stage 4. Health status showed a relationship to spirometric severity with worsening for each stage although (as expected) with a wide range and major group overlap as reported from the larger ADAPT database[5] and other studies of non-AATD COPD.[6] Importantly, the patients in GOLD stage 4 had a median total SGRQ score 62.0 (IQR 36.5–67.3) which was similar to that for 51 stage 4 patients recruited previously to the ADAPT database (extracted from Stockley et al[5]) with a median value of 59.1 (IQR 51.8–72.0). This suggests that our previous data were not influenced by acquisition bias

as a result of travel to a single centre being influenced by distance and/or patient health. In addition, it indicates that GOLD stage4 patients can be readily recruited from multiple sites for research, audit and appropriate clinical trial purposes.

Relationship of lung densitometry (the most direct measure of pathological emphysema) to objective parameters of lung function was similar to that seen in other studies[7 8] but better for gas transfer than spirometry (consistent with the former being a more direct measure of alveolar dysfunction as seen in emphysema) and certainly better than the weak relationship to air trapping (RV/TLC) and the subjective health status measures. COPD, even in AATD, is a multifaceted condition although classically characterised in AATD by the presence of panacinar emphysema localised to the lung bases as the dominant pathological phenotype. Lung densitometry is the most direct measure of this process and its severity is demonstrated pathologically.[9] All other measures are indirect surrogate markers of emphysema, and although densitometry in AATD relates to exercise capacity,[10] spirometry and gas transfer[11] and health status,[12] it remains the best independent predictor of mortality in this disease.[13] In addition, it is the most sensitive measure of emphysema progression in AATD[14] and hence its support as the primary outcome in clinical trials enabling such studies to be sufficiently powered for such a rare disease. CT has not been routinely undertaken as part of management of AATD patients largely on the basis of cost and radiation exposure. However, the radiation exposure protocol for densitometry is less than that for high-resolution

scans and is specific for the pathological phenotype of the COPD in AATD (namely the emphysema). Indeed the recent European Respiratory Society (ERS) strategy document on AATD has confirmed it as part of the assessment for all such patients and that referral to centres with a major interest in AATD should be included in patient management.[15] The relationships of CT densitometry to more recognised parameters of COPD severity and mortality are no worse and certainly better (specifically mortality) than some of the more conventional markers of the disease although still only weakly related to the subjective interpretation of the patients' health.

The study was successful in enabling collaboration between academic centres by providing recruitment and in-depth phenotyping of a group of patients with AATD and a wide range of physiological impairment to an agreed standard. All subjects also had plasma taken and stored for the development and validation of assays to monitor serine proteinase footprints[16] and other disease-specific biomarkers. In addition, the network provides baseline cohort data for long-term follow-up and importantly a well-characterised patient database for collaboration with industry (one of the key aims for the NIHR programme). Furthermore, it conforms to the key elements outlined by the ERS strategy document for the management of AATD[15]; namely, establishment of registries, comprehensive assessment and review by specialists with an interest in AATD. It also provides the background for linking with the European Union rare diseases collaborative and the European Alpha-1 Research Collaboration.[17]

Nevertheless, the study has some limitations. Several of the patients had undergone CT scanning as part of their routine clinical assessment prior to the establishment of the collaborative and hence it was not deemed ethical to repeat the procedure in the short term for densitometry analysis as part of a phenotyping study. However, the parameters are set for all future scans in this patient population both at baseline and for monitoring progressions. The documentation of health status was not perfectly consistent although either the SGRQ or CAT was performed in all subjects. Only 85% had the SGRQ documented, whereas 95% had the CAT documented, which may reflect the latter's ease of administration and practical clinical utility.[4] Our analysis reported here only included those patients in whom both had been measured. Currently, only the SGRQ has been accepted as a validated tool for patient-reported outcomes in clinical trials,[18] whereas the CAT has become an accepted measure in patient management as supported by GOLD[4] and is often also reported in the clinical studies and trials. However, such measures of health status are best used to indicate a noticeable change from baseline for therapies that provide an initial and detectable impacts such as bronchodilators and not therapies that modify slow progression of chronic diseases such as COPD and emphysema.[5] Therefore, although densitometry relates to both $FEV_1$ and health status (though poorly for the latter), evidence of progression and response to treatment in AATD should currently depend on CT scans as the most direct and sensitive marker of emphysema and its progression as confirmed recently in a placebo-controlled trial of the benefit of augmentation therapy.[19]

In summary, the project successfully recruited to 94% of target in the 1-year time frame and the majority underwent deep phenotyping to a set standard for future patient care and assessment consistent with the view of the ERS strategy document for AATD.[15] The collaboration and expertise of participants provide a firm structure for future management of AATD in the UK. Importantly, it provides validation that previous patient assessment as part of the ADAPT programme reflects patients across the spectrum of disease severity and not a patient acquisition bias related to distance or travel. Furthermore, it demonstrates an intercalated network that can harmonise activity and research outcomes using standard methodologies. These points give reassurance that patient monitoring and care can be provided at multiple sites to the same standard, factors that are both critical in patient convenience and reassurance, as expressed by patients in the recent ERS AATD strategy document.[15] Importantly, it provides validation of CT densitometry for characterisation of the pathological nature and severity of the disease and places this in perspective with other measures routinely used for monitoring. Because of its specificity, it should become an essential part of AATD assessment.[15] Finally the collaborative provides an essential, ready-made resource for the delivery of research, clinical studies and trials especially for the pharmaceutical industry and a UK resource for patient/specialist research collaboration.

**Author affiliations**
[1]College of Medical and Dental Sciences, Institute of Inflammation and Ageing, Centre for Translational Inflammation Research, Queen Elizabeth Hospital, University of Birmingham, Birmingham, UK
[2]University Hospitals Birmingham NHS Foundation Trust, Queen Elizabeth Hospital Birmingham, Mindelsohn Way, Edgbaston, Birmingham, UK
[3]Department of Respiratory Medicine, NIHR Nottingham BRC respiratory theme, School of Medicine, The University of Nottingham, City Hospital Campus, Nottingham, UK
[4]National Heart and Lung Institute, Imperial College, Royal Brompton Hospital Campus, London, UK
[5]Department of Medicine, Cambridge NIHR BRC, University of Cambridge, Leicester, UK
[6]NIHR Leicester Biomedical Research Centre – Respiratory, Institute for Lung Health, University of Leicester, Glenfield Hospital, Leicester, UK
[7]Respiratory BRU, University of Southampton, Southampton, UK
[8]Respiratory, Royal Free Campus, London, UK
[9]King's College London, Guy's Hospital Site, Great Maze Pond, London

**Acknowledgements** We would like to thank Anita Pye, the study coordinator for her assistance and support and Mr J Beggs the patient representative on the Steering committee.

**Author Contributions** The project concept was developed by RS; JS, CB, NH, JRH, BG, RM, MS, TW and RS established the network as part of the NIHR rare diseases call, collated clinical, physiological and radiological data across each contributing organisation. DC analysed the quantitative CT scans and the related data. DC drafted and refined the manuscript with RS. All authors reviewed, modified and approved the final manuscript.

**Funding** The research and network were funded by the National Institute for Health Research(NIHR) rare diseases TRC and coordinated from the NIHR/Wellcome Trust Birmingham Clinical Research Facility. NIHR Nottingham BRC respiratory theme, Cambridge NIHR BRC, NIHR Leicester BRC—respiratory theme, Respiratory BRU Southampton, NIHR Respiratory BRU at Royal Brompton and Harefield NHS Foundation Trust and Imperial College, London.

**Competing interests** RS sits on several advisory boards for pharmaceutical companies with/or developing, treatments for AATD and NSH is currently recruiting to a phase 2 study in these patients sponsored by Mereo Biopharma.

**Patient consent for publication** Not required.

**Ethics approval** All patients gave informed consent and the study was approved for all sites by the local ethics committee (West Midlands—South Birmingham Research Ethics Committee, Local Research Ethics Committee (LREC) 3359).

**Provenance and peer review** Not commissioned; externally peer reviewed.

**Data availability statement** No additional data are available.

**ORCID iDs**
John R Hurst http://orcid.org/0000-0002-7246-6040
Robert A Stockley http://orcid.org/0000-0003-3726-1207

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
