## [Reviewer comments · BMJ Open]

ARTICLE DETAILS

TITLE (PROVISIONAL)	The relationship of CT densitometry to Lung Physiological parameters and Health Status in Alpha-1 Antitrypsin deficiency; Initial report of a centralised data base of the NIHR Rare diseases Translational Research Collaborative.
AUTHORS	Crossley, Diana; Stockley, James; Bolton, Charlotte; Hopkinson, Nicholas; Mahadeva, Ravi; Steiner, michael; Wilkinson, Tom; Hurst, John; Gooptu, Bibek; Stockley, Robert;

VERSION 1 – REVIEW

REVIEWER	Per Wollmer Lund University Sweden
REVIEW RETURNED	22-Jan-2020

GENERAL COMMENTS	This manuscript reports an effort to establish a registry of subjects with AATD in the United Kingdom. The registry includes clinical information in the form of questionnaires, pulmonary function tests and chest CT. The authors set a target to include 200 subjects, whereof half from the central facility, in one year. Data were collected for 187 subjects and complete data (?) from 147 subjects. Mainly chest CT was missing. The authors present descriptive information and correlations between questionnaires, lung function and CT. Major comments 1. There is very little new, scientific information in this manuscript. There are several registries of subjects with AATD in various countries and the one presented is neither very different nor very large. The information mainly has a national interest. I doubt that many international readers will find it very exciting.2. The description of the registry is insufficient. It is not clear by which criteria subjects were included. In the discussion, the authors state “we wished to include those with very severe disease (GOLD stage 4) as quantified by baseline FEV1”. There thus appears to be some form of stratification, which should be described. Since the object was to include half of the subjects from hospitals other than the central facility, the proportion should be reported. The pulmonary function tests need to be specified. What is meant by “full post bronchodilator lung function testing”? Which equipment was used in the different centres? Which standards were applied? What reference values were used? A more general question concerns quality control. Providing a “standard protocol” is all very well, but how was adherence monitored?
---

	3. Chest CT was not repeated if performed within two years of inclusion. This means that a proportion of the examinations were analysed retrospectively. This proportion should be reported. Were all these examinations performed according to the standard protocol? Minor comment 1. Numerical information should be reported in text or tables, not in both.
--	--

REVIEWER	Igor Barjaktarevic UCLA, Los Angeles, CA, USA
REVIEW RETURNED	10-Mar-2020

GENERAL COMMENTS	The manuscript of Crossley et al. reports results of a multicenter collaboration focused on AATD and proves the need to establish registries and apply comprehensive approach to AATD patients. It offers some validation of CT densitometry for characterization of the pathological nature and severity of the disease. The study fulfilled one of its goals to allow for the recruitment of AATD patients with more advanced disease, capturing more than quarter of all enrolled subjects to represent GOLD stage IV obstruction. It is well written by the group of experts in the field and adds on the evidence in published literature about the disease and the usefulness of imaging in AATD. One of my major concerns is that the paper does not offer much in terms of novelty. Wisely chosen imaging metrics, PD15 and -950HU have been previously reported sensitive and probably most adequate for evaluating the lung function. In this study, we do see similar findings which show relatively good correlations with FEV1 and Kco. Showing that these metrics also correlate (not very strongly) with metrics reflecting symptoms and QoL, raises some concern about the adequacy of the univariate analysis; there is concern for collinearity with FEV1, and this relationship should maybe be tested in a multivariate analysis adjusted for FEV1 in order to offer more adequate answers. If not decided to explore this option given small N, I recommend this to be discussed. With the concern about confounding effect of FEV1 on the relationship between PD15/voxels and CAT/SGRQ, one can critically question what imaging adds on the the FEV1/DLCO testing. How will the authors – based on these data - justify the use of expensive and harmful diagnostic tool in this population if it does not relate to the outcomes (like shown in reference #13)? I think the second to the last sentence: “Because of its specificity, (missing coma) it should become an essential part of AATD assessment” in conclusion is not sufficient without further explanation. We probably cannot draw the conclusion that imaging is a good end point for studies based on correlations with some baseline features. I think the authors may have a chance to
--

elaborate more on the role of imaging as a potential one-stop shop that gives anatomic insight into diseased lungs, offers the data otherwise provided by complete PFT, correlates with subjective features of the disease (as shown here) and allows for longitudinal follow up (as shown elsewhere); by doing this, the authors may further emphasize the strength of their report which offer a piece which further solidifies the statement above.

Some concern about the ability of the baseline and potential longitudinal lung function relationship with these imaging metrics is coming from the fact that findings here are consistent with previous reports showing good (even better than here) correlation between FEV₁ and PD15 but the correlation between the decline in FEV₁ and progression of PD15 was very weak (Thomsen LH et al, COPD, 2015). While this study may not have this data to analyze, I think it is important to include this problem in the discussion part.

I also wonder if there may be an option to make these results more granular by showing that these correlations between spiro/Kco and PD15/Voxels remain similar in all stages of COPD and are applicable throughout the wide spectrum of disease. Thus, analysis of the correlations of PD15 and voxel index between people with mild/moderate vs severe/very severe obstruction may offer some insight of how universally useful these imaging metrics can be. Again, having some (baseline) exacerbation data would certainly be helpful as - in case the data is available, it could allow the authors to look at the correlation of GOLD groups (A-B-C-D, rather than stages) and allow for evaluation of association of CT measures with exacerbations, and more importantly the extent of the disease (measured by symptoms x AE).

RV/TLC may be better presented by % *ref* rather than *absolute percentage* (Table 1)

Finally, the cohort of the patients enrolled consists of sick people, as reflected in abnormal CAT scores even in the ones without the obstruction. One can speculate that the information about comorbidities, BMI or exacerbations could help better characterize this cohort and help reduce possible selection bias which was favoring compliant, well followed (on replacement therapy? No data offered) cohort of airway-diseased subjects, disabling disease-wide conclusions about the utility of imaging in disease characterization. This should be further discussed.

Less relevant flaws include slightly suboptimal collection of the data – besides the incomplete data on CAT and SGRQ (especially in the stage IV), current smoking history (is there any active smoker in the cohort?), exacerbations data are missing as well, as

	well as almost quarter of subjects missing protocolized CT. The authors do address some of these, but certainly not all. Would add a sentence of two rationalizing why 910 or 856, or lung density were not used here and why the two are chosen. In summary, this report does bring up a great point in terms of advocating for collaborative efforts in AATD and demonstrating a successful network created in the UK, and it advocates for a lung densitometry as a clinical end-point which deserves further solidification, all important issues worth reporting. Possible inclusion of the longitudinal data in the followed cohort and relationship of imaging metrics to exacerbations and FEV1 decline would certainly offer more relevance to the study and I can hope that the registry will offer the chance to have these analyses in the future. I'd say that without a little of additional "boost" to the findings in this manuscript, I find that this paper is good but without too many exciting or novel results. It can be published, although I would recommend considering some of my suggestions above to be added. Minor remarks: Interpunction is inadequate, a lot of comas are missing – some (of many) examples include Page 12, first paragraph: "All other measures are indirect (double space here) surrogate markers of this process and, although densitometry..."; second paragraph: "Furthermore, it confirms...", or third paragradh – "Nevertheless, the study..." or "However, the parameters...". There are many more examples. Consider using database as a single word (webster's dictionary) Not sure FEV1 and FVC in absolute values (L) are necessary in the Table 1. Express RV/TLC as % ref
--	---

VERSION 1 – AUTHOR RESPONSE

BMJ open response to reviewers.

The minor issues raised by the editor have been addressed:- namely attempting a more structured title and removing the Article summary heading

Reviewer 1.

This manuscript reports an effort to establish a registry of subjects with AATD in the United Kingdom. The registry includes clinical information in the form of questionnaires, pulmonary function tests and

chest CT. The authors set a target to include 200 subjects, whereof half from the central facility, in one year. Data were collected for 187 subjects and complete data (?) from 147 subjects. Mainly chest CT was missing. The authors present descriptive information and correlations between questionnaires, lung function and CT.

Major comments

1. There is very little new, scientific information in this manuscript. There are several registries of subjects with AATD in various countries and the one presented is neither very different nor very large. The information mainly has a national interest. I doubt that many international readers will find it very exciting.

We agree that other registries have been developed. However this represents a network cohort working to a common data base providing a linked system for clinical trials in untreated patients and successfully recruited 187 patients over 12 months using a standard protocol. The collection of data from outlying sites confirms that severe COPD patients assessed at a single central site is also representative of data from patients (n=) recruited from sites near to the patients base overcoming potential acquisition bias. It also included CT densitometry together with more extensive lung function than is routinely collected in other registries according to a defined standard (see below)

2. The description of the registry is insufficient. It is not clear by which criteria subjects were included. This was stated based on spirometric criteria and severity of airflow obstruction as staged by GOLD . We have adjusted the wording to make this more clear. Recruitment was sequential based on consent

In the discussion, the authors state “we wished to include those with very severe disease (GOLD stage 4) as quantified by baseline FEV₁”. There thus appears to be some form of stratification, which should be described.

This was described and has been clarified

Since the object was to include half of the subjects from hospitals other than the central facility, the proportion should be reported.

IT WAS Restated and numbers included

The pulmonary function tests need to be specified. What is meant by “full post bronchodilator lung function testing”?

Clarified as “according to ARTP/BTS guidelines for quality control”.

Which equipment was used in the different centres? This is not felt necessary as all were academic centres of excellence as defined by funding by NIHR Which standards were applied?ARTP /BTS standards apply across the UK academic centres. This has been mentioned in the text and a reference can be given if felt necessary

What reference values were used?European Coal and steel
.This was quoted in the original article page 7 reference 3.

A more general question concerns quality control. Providing a “standard protocol” is all very well, but how was adherence monitored?

All centres are validated for adherence to the NationalARTP/BTS and why only academic centres were used to delivering clinical trials protocols were included (as stipulated by the NIHR before funding approval)

3. Chest CT was not repeated if performed within two years of inclusion. This means that a proportion of the examinations were analysed retrospectively. This proportion should be reported. Were all these examinations performed according to the standard protocol?

No the previous scans were not obtained and analysed. The protocol parameters outlined in the methods were specific for analysis We only report the prospective scans that were collected per protocol. This is a unique aspect of the current registry.

Minor comment

1. Numerical information should be reported in text or tables, not in both .

Revisited

Reviewer: 2

Reviewer Name: Igor Barjaktarevic

Institution and Country: UCLA, Los Angeles, CA, USA

Please state any competing interests or state 'None declared': None declared

Please leave your comments for the authors below

Great network, great idea and intent to further solidify imaging as a relevant tool in AATD, but relatively modest results without too much novelty and creativity. Overall, still a good simple paper. Thank you for this support

VERSION 2 – REVIEW

REVIEWER	Igor Barjaktarevic UCLA, USA
REVIEW RETURNED	29-Apr-2020
GENERAL COMMENTS	Limited response by the authors to this reviewer's suggestions, and the revised version is somewhat improved in comparison to the previously submitted version; yet, I have no objections to have this manuscript accepted. Revised version has a lot of syntax errors, needs corrections

VERSION 2 – AUTHOR RESPONSE

My apologies over the STROBE check list as I thought this was just for guidance and I have enclosed a document with this.

Secondly I have obtained all lung function equipment details from all centres although this took some time because of lockdown. This is titled as supplement 2 as I think it should just be added to the original supplement that remains unchanged. I have referred to this in the methods.

Thirdly I have added a further copy of the main manuscript that has been updated and reviewed this with reviewer 2 comments in mind